# AdipoRon as a Novel Therapeutic Agent for Depression: A Comprehensive Review of Preclinical Evidence

**DOI:** 10.3390/biomedicines13081867

**Published:** 2025-07-31

**Authors:** Lucas Fornari Laurindo, Victória Dogani Rodrigues, Rodrigo Haber Mellen, Rafael Santos de Argollo Haber, Vitor Engrácia Valenti, Lívia Fornari Laurindo, Eduardo Federighi Baisi Chagas, Camila Marcondes de Oliveira, Rosa Direito, Maria Angélica Miglino, Sandra Maria Barbalho

**Affiliations:** 1Postgraduate Program in Structural and Functional Interactions in Rehabilitation, School of Medicine, Universidade de Marília (UNIMAR), Marília 17525-902, São Paulo, Brazil; 2Department of Biochemistry and Pharmacology, School of Medicine, Universidade de Marília (UNIMAR), Marília 17525-902, São Paulo, Brazil; 3Department of Biochemistry and Pharmacology, School of Medicine, Faculdade de Medicina de Marília (FAMEMA), Marília 17519-030, São Paulo, Brazil; 4Autonomic Nervous System Center, School of Philosophy and Sciences, São Paulo State University, Marília 17525-902, São Paulo, Brazil; 5Department of Biochemistry and Pharmacology, School of Medicine, Faculdade de Medicina de São José do Rio Preto (FAMERP), São José do Rio Preto 15090-000, São Paulo, Brazil; 6Laboratory of Systems Integration Pharmacology, Clinical and Regulatory Science, Research Institute for Medicines, Universidade de Lisboa (iMed.ULisboa), Av. Prof. Gama Pinto, 1649-003 Lisbon, Portugal; 7Department of Biochemistry and Nutrition, School of Food and Technology of Marília (FATEC), Marília 17500-000, São Paulo, Brazil; 8Department of Research, Research Coordination Center, UNIMAR Charitable Hospital, Universidade de Marília (UNIMAR), Marília 17525-902, São Paulo, Brazil

**Keywords:** AdipoRon, adiponectin replacement therapy, adiponectin, AdipoR1, AdipoR2, depression, central nervous system

## Abstract

**Background/Objectives:** Depression is a mood disorder that causes persistent sadness and loss of interest, and its etiology involves a condition known as hypoadiponectinemia, which is prevalent in depressive individuals compared with healthy individuals and causes neuroinflammation. The use of intact adiponectin protein to target neuroinflammation in depressive moods is complex due to the difficulties associated with using the intact protein. AdipoRon, a synthetic oral adiponectin receptor agonist that targets the AdipoR1 and AdipoR2 receptors for adiponectin, has emerged in this context. Its most prominent effects include reduced inflammation and the attenuation of oxidative stress. To the best of our knowledge, no comprehensive review has addressed these results so far. To fill this literature gap, we present a comprehensive review examining the effectiveness of AdipoRon in treating depression. **Methods:** Only preclinical models are included due to the absence of clinical studies. **Results:** Analyzing the included studies shows that AdipoRon demonstrates contrasting effects against depression. However, most of the evidence underscores AdipoRon-based adiponectin replacement therapies as potential candidates for future treatment against this critical psychiatric condition due to their anti-neuroinflammatory potential, ultimately inhibiting several neuroinflammatory pathways. **Conclusions:** Future research endeavors must address several limitations due to the heterogeneity of the studies’ methodologies and results.

## 1. Introduction

Depression is a mood disorder that causes persistent sadness and loss of interest. Although the Diagnostic Statistical Manual of Mental Disorders (DSM-5) characterizes the existence of a variety of entities that could be classified into the spectrum of depressive disorders, the standard features among all are sadness, irritable mood, and emptiness accompanied by somatic and cognitive damages [1]. Depression raises concerns mainly because it significantly affects the capacity to function and quality of life of those affected. Additionally, depression is well known as a potential risk factor for the development of not only mild cognitive impairment, but also dementia, including Alzheimer’s disease [2]. Epidemiologically, it is estimated that 21 million Americans had at least one major depressive episode during 2020, which represents 8.4% of the United States population [3]. Depression may affect mothers during the perinatal period and lead to an elevated risk of attention deficit and hyperactivity disorder symptoms in offspring [4]. Depression may strongly affect cancer patients [5], as well as their caregivers [6], even today, when depression has been indicated as a highly feasible target for renewed clinical attention due to its relationship with systemic inflammation and mortality within cancer treatment [7]. Significantly, depression also heavily affects older people. A recent meta-analysis revealed that the prevalence of depression, anxiety, and stress in the elderly population is high. Its overall estimates for prevalence were 19.2% for depression, 16.5% for anxiety, and 13.9% for stress, and this is because depression, anxiety, and stress are often indissociable [8]. On the other hand, in the pediatric population, from 2004 to 2019, pooled prevalence estimates were 0.71% for major depressive disorder, 0.30% for dysthymia, and 1.60% for disruptive mood dysregulation disorder [9]. A meta-analysis also indicated that adolescents who experience depression develop a higher incidence of suicidal behavior in adulthood [10].

Although the etiology of major depressive disorders is multifactorial, studies have shown that a condition known as hypoadiponectinemia is prevalent in depressive individuals compared with healthy ones [11]. In preclinical studies, adiponectin has been shown to promote adult neurogenesis, enhance synaptic plasticity in the hippocampus, and facilitate dendritic and spine remodeling, ultimately leading to antidepressant effects in many mouse models of depression [12]. These results are prominent mainly because of adiponectin’s anti-neuroinflammatory potential against several brain disorders [13]. Despite technological, nanotechnological, and nano-conjugation advancements, using the intact adiponectin protein remains a challenge, and converting the complete protein into a viable drug candidate remains impossible for two reasons. Firstly, there is a diverse range of adiponectin structures, which hinders the attainment of consistently reproducible outcomes in laboratory settings (in vitro) and living organisms (in vivo). Secondly, the C-terminal domain of adiponectin exhibits significant insolubility. Due to these reasons, translating viable adiponectin-based therapies into clinical practice remains impractical, and substitutes have begun to be developed.

AdipoRon (Figure 1), an adiponectin receptor agonist, is a synthetic molecule with significant potential for treating various diseases and conditions. The molecule has the potential to effectively mimic adiponectin signaling by activating both the adiponectin receptor 1 (AdipoR1) and adiponectin receptor 2 (AdipoR2) receptors [14]. AdipoRon is the most extensively studied adiponectin receptor agonist for adiponectin replacement therapies. AdipoRon can be administered orally, and numerous positive effects have been associated with its use. Its main effects are the reduction in inflammation [15], enhancement of lipid metabolism [16], improvement of mitochondrial function [17], and attenuation of oxidative stress [18]. Published reviews have highlighted AdipoRon’s potential against metabolic-associated fatty liver disease and nonalcoholic steatohepatitis [19], hormone-related cancers such as pancreatic and gynecological cancers and osteosarcoma [20], and diabetic nephropathy and cardiomyopathy prevention and intervention [21]. Reviews have also been published highlighting AdipoRon’s potential in modulating the physiology and response to stress conditions of ovarian granulosa cells [22] and bone metabolism and repair [23]. More recently, a systematic review and meta-analysis assessed the effects of AdipoRon administration in preclinical models of Alzheimer’s disease, demonstrating AdipoRon’s beneficial impacts on modulating neuroinflammation and reducing dementia symptoms and pathways in various preclinical experiments [24]. However, no clinical studies or randomized trials were performed using AdipoRon to treat diseases directly in human subjects.

In the past, a single review by Formolo et al. [12] addressed the potential of AdipoRon in treating depression. However, the evidence from this previous review is obsolete. Since then, numerous other works have been published, and the need for an updated comprehensive review has increased. Given the above scenario, this is the first updated systematic review to comprehensively assess AdipoRon’s effectiveness against depression in six years, focusing on preclinical studies due to the absence of pertinent clinical trials. To emphasize the significance of the present work, we employed the PICO framework—Population, Intervention, Comparison, and Outcomes—in constructing this review. The included studies were reviewed for intervention, control, endpoints, results, and conclusions to maintain consistency and ensure accurate data interpretation. With the publication of this comprehensive review, our primary objective is to highlight a new therapeutic option for depression and encourage the conduct of translational studies in clinical trials to translate preclinical findings into robust clinical approaches.

## 2. Materials and Methods

### 2.1. Database Search

A comprehensive literature review was conducted to evaluate the effects of AdipoRon-based adiponectin replacement therapy against depression. Multiple databases, including PubMed, Scopus, Web of Science, and Google Scholar, were comprehensively searched using the PICO framework (Population, Intervention, Comparison, and Outcome). Keywords included “AdipoRon”, “adiponectin replacement therapy”, “depression”, “in vitro”, “in vivo”, “neuroinflammation”, and “cognitive function.” Boolean operators, such as AND, OR, and NOT, were used to refine the search results; for example, “AdipoRon AND depression.” Filters such as publication date (within the last decade), language (English), and document type (original research articles) were applied to ensure the relevance and quality of the included studies. Our search strategy aimed to capture a broad range of studies that focused on using AdipoRon-based therapies against depression. This framework ensured a comprehensive review of the available evidence.

Our PICO framework utilization is depicted below.

(P) We included preclinical models of depression (including preclinical cellular and animal studies).

(I) We included studies that utilized AdipoRon-based adiponectin replacement therapies against depression.

(C) We included studies that incorporated control groups (e.g., untreated models or those given placebo/vehicle). We considered adding studies also comparing AdipoRon with other antidepressant treatments if available, though the focus is primarily on AdipoRon vs. no intervention.

(O) We included studies analyzing depression-like behaviors, neuroinflammation, relevant molecular pathways, cognitive function, and the overall therapeutic effects of AdipoRon, including those metabolically related to its systemic effects.

(Question) In preclinical models of depression (P), does treatment with AdipoRon (I), compared to no treatment or placebo (C), improve depression-related outcomes such as behavioral symptoms, neuroinflammation, and cognitive function (O)?

### 2.2. Inclusion Criteria

Due to the absence of clinical trials, the inclusion criteria for the studies were explicitly focused on preclinical research. This included studies involving either in vitro or in vivo models of depression that featured experimental groups treated with AdipoRon. We required that the studies report outcomes related to depression-like behaviors, neuroinflammation, relevant molecular pathways, and other outcomes to comprehensively evaluate AdipoRon’s effects against depression. Studies were required to have clear and robust experimental designs and accurately report their results. They had to be published in peer-reviewed journals and provide detailed information on the intervention, control, endpoints, results, and conclusions. Only articles published in the last decade were considered to ensure the relevance of the findings to current research and advancements. Additionally, only studies published in English were included to maintain consistency and ensure accurate interpretation of the data. Although the language criterion might be considered a limitation, a systematic review report has demonstrated no evidence of systematic bias due to the addition of language restrictions [25].

### 2.3. Exclusion Criteria

The exclusion criteria included reviews, meta-analyses, non-experimental papers, studies not involving AdipoRon as an intervention, and those not based on models of depression. Studies with inadequate experimental designs or unclear reporting of results were also excluded. Studies that had been published but were retracted were also excluded to maintain the highest level of evidence.

### 2.4. Data Extraction

Data extraction was performed using a standardized form to capture essential information on experimental models, AdipoRon treatment details (concentration and duration), outcomes, and limitations. Data extraction was performed by two authors, L.F.L. and S.M.B. Any disagreements were resolved by a third author, V.D.R. Nevertheless, no disagreements arose between the authors during the process of searching and selecting the appropriate studies that met our inclusion criteria.

### 2.5. Quality Assessment

The quality of the studies was assessed based on their experimental design, sample size, and clarity of result reporting, following established guidelines for scientific rigor and using SYRCLE’s risk of bias tool for animal studies [26]. Initially, the methodological rigor of each study was evaluated based on the design and execution of experiments, the appropriateness of models and controls, and the clarity with which interventions and endpoints were defined. The sample size of the included studies was examined to ensure that it was adequate for drawing reliable conclusions, and the comprehensiveness of data reporting was assessed to verify that the results were presented with sufficient detail. Potential biases were identified, including variability in experimental conditions and differences between short-term and long-term outcomes. The studies’ ability to translate preclinical findings into potential clinical implications received particular attention. A qualitative data synthesis was conducted to summarize the effects of AdipoRon on depression models, identify expected outcomes, and discuss limitations. The findings were organized to highlight the therapeutic effects of AdipoRon, its mechanisms of action, and its potential as a treatment for depression. By systematically applying these criteria, we aimed to ensure a thorough and objective evaluation of the studies’ overall quality and relevance for advancing AdipoRon-based therapies for depression. This review also aimed to identify gaps in current research and suggest directions for future studies and clinical applications.

## 3. Results


*Literature Search Report*


Initially, we identified 125 records from PubMed (*n* = 75), Scopus (*n* = 30), Web of Science (*n* = 10), and Google Scholar (*n* = 10), as well as 20 records from registers. After a validation screening, 50 records were excluded because they were duplicates, 26 were marked as ineligible by automated tools, and 18 were removed for other reasons (17 studies were not in English, and 1 study was not published within the last ten years). After this initial validation, 51 records were then screened for inclusion. Forty-one studies were excluded at this stage because they were reviews and other non-experimental papers. Then, ten studies were sought for retrieval, and fortunately, all ten studies were retrieved. Finally, the ten studies were assessed for their eligibility, and seven were included in the final analysis. At this stage, three studies were excluded based on the following criteria: two studies did not involve AdipoRon-based adiponectin replacement therapies, and one study did not include models of depression. Table 1 summarizes the included studies on the impact of AdipoRon on depression. Figure 2 illustrates the literature search report.

The seven included studies assessed AdipoRon’s effects on mouse models of depression. These were related to the 6-hydroxydopamine (6-OHDA)-induced Parkinson’s Disease (PD) rat model [27], depression-like behaviors in mice [28,29,30,31], lipopolysaccharide (LPS)-induced depression-like behaviors in mice [32], and long-term corticosterone treatment for depression in male mice [33]. Zhao et al. demonstrated the antidepressant and neurostimulator effects of AdipoRon in diabetic and depressive mice via enhanced anti-inflammatory effects. Azizifar et al. demonstrated and assessed intranasal-administered AdipoRon’s effects on depressive symptoms. By decreasing inflammation, Liu et al. demonstrated AdipoRon’s effects in improving neuroprotection and mitochondrial function. Formolo et al. assessed depressive symptoms and AdipoRon treatment, but did not establish the molecular mechanisms involved. Li et al.’s experiments were the only ones to determine the mechanisms of AdipoRon against LPS-involved depression. You et al.’s experiments demonstrated that AdipoRon rescued neurogenesis and ameliorated cognitive dysfunction. Furthermore, Nicolas et al. modulated several molecular mechanisms with AdipoRon treatment, including inflammatory and neuroprotective pathways. Key differences included the models tested, the signaling pathways involved, methodological limitations (such as the omission of pharmacokinetic parameters), and the future research directions that their results may suggest.

**Table 1 biomedicines-13-01867-t001:** Impact of AdipoRon on depression: insights into models, interventions, and mechanisms.

Model Type	Cell Line or Animal	Intervention and AdipoRon Administration	Antidepressant Effects	Pathways	References
**In vivo *experiments*** Depression-like behavior in STZ + high-fat-diet-induced diabetic rat model	**In vivo *experiments*** C57BL/6N mice	**In vivo *experiments*** **(i.p., injected for two weeks)** AdipoRon 20 mg/kg	AdipoRon at a dose of 20 mg/kg significantly improved depression-like behaviors in mice, enhancing sucrose consumption in the sucrose preference test, reducing immobility time in the swimming experiment, and increasing total distance and cross times in movement experiments.	AdipoRon inhibited hippocampal cell apoptosis, increased synapses in the prefrontal cortex and hippocampus, and increased dendritic spine neuronal density in the CA1 region. The molecular pathways included anti-inflammatory effects via NLRP3 inhibition, reduction in ASC and IL-1β levels, increased p-AMPK/AMPK ratio, and decreased p-mTOR/mTOR expression in the treated mouse brains.	[31]
**In vivo *experiments*** 6-OHDA-induced PD rat model	**In vivo *experiments*** Wistar rats	**In vivo *experiments*** **(intranasal, injected for three weeks)** AdipoRon 0.1 µg/rat, AdipoRon 1 µg/rat, and AdipoRon 10 µg/rat	AdipoRon at doses of 1 and 10 µg/rat for three consecutive weeks was associated with anxiolytic and antidepressant effects in many behavioral tests, such as the OF, EPM, and forced swimming tests. AdipoRon also lowered corticosterone and inflammasome levels in the treated rats.	AdipoRon mitigated anxious and depressive-like behaviors in the rat model of PD by principally modulating the AMPK/Sirt-1 (↓NLRP3, IL-1β, CAS-1, and ↑Sirt-1) signaling pathway and blocking the NLRP3 inflammasome.	[27]
**In vitro *experiments*** Microglial and hippocampal cells **In vivo *experiments*** CUMS	**In vitro *experiments*** BV2 and HT22 cells **In vivo *experiments*** C57BL/6J mice	**In vitro *experiments*** BV2 cells were pretreated with AdipoRon 10, 20, and 40 μM for two hours, and HT22 cells were cultured with the BV2 cell-conditioned medium **In vivo *experiments*** **(i.p., injected for two weeks)** AdipoRon 10 mg/kg, AdipoRon 20 mg/kg, and AdipoRon 40 mg/kg	AdipoRon protected hippocampal neurons cultured with activated microglia, decreased mtROS accumulation, and promoted mitophagy in vitro, which increased the clearance of damaged mitochondria. Additionally, AdipoRon ameliorated depression-like behaviors in vivo.	AdipoRon mitigated NLRP3 inflammasome activation in the microglia by improving mitophagy.	[28]
**In vivo *experiments*** AAV (↑shRNA targeting NMDA subunits Glu2NA and Glu2NB)	**In vivo *experiments*** C57BL/6J and CamKIIα-Cre mice	**In vivo *experiments*** **(i.p., injected for seven days)** AdipoRon 20 mg/kg	AdipoRon promoted antidepressant and anxiolytic-like effects even under the knockdown of the NMDA receptor subunits GluN2A and GluN2B in the ventral hippocampus of the treated mice. AdipoRon reduced BDNF levels, long-term potentiation of the perforant path, and neuronal activation in the ventral dentate gyrus.	-	[29]
**In vivo *experiments*** LPS-induced depression-like model	**In vivo *experiments*** C57BL/6 APN KO mice	**In vivo *experiments*** **(i.g, injected for seven days)** AdipoRon 50 mg/kg	AdipoRon was found to abolish the antidepressant-like behaviors presented by LPS-treated APN KO mice, increasing immobility and decreasing sucrose preference. However, AdipoRon improved the redox status of the treated mice.	The suggested pathway involved possibly an anti-neuroinflammatory intervention based on the modulation of AdipoRon and BDNF signaling.	[32]
**In vivo *experiments*** CDAD	**In vivo *experiments*** C57BL/6J mice	**In vivo *experiments*** **(i.v. cannulation and i.c.v. injection)** AdipoRon 1 µL/1 mM	AdipoRon promoted hippocampal neurogenesis and improved cognitive dysfunction associated with depression.	AdipoRon increased the genetic expression of NICD, ADAM10, Hes1, Hes5, Hey1, and Heyl and upregulated Notch1 signaling. Additionally, it increased the number of Ki67- and DCX-positive cells. AdipoRon may have upregulated the expression of ADAM10 and Notch1 through PPARα and JNK, respectively.	[30]
**In vivo *experiments*** Long-term corticosterone treatment	**In vivo *experiments*** C57BL/6J mice	**In vivo *experiments*** **(i.p., injected for three weeks)** AdipoRon 1 mg/kg	AdipoRon positively impacted a depression-like state by blocking corticosterone-induced depression onset. It exerted pleiotropic effects, modulating hippocampal neurogenesis, tryptophan metabolic pathways, neuroinflammation, and serotonergic transmission. AdipoRon mitigated depression-like behaviors.	AdipoRon modulated IL-1β, IL-6, and TNF-α, restored the physiological expression of IDO and KAT in various brain regions, increased the release and turnover of serotonin, and restored physiological levels of critical neurotrophic factors such as BDNF, VEGF-α, IGF-1, and NGF.	[33]

**Abbreviations:** 6-OHDA, 6-hydroxydopamine; ADAM10, A disintegrin and metalloprotease 10; ASC, apoptosis-associated speck-like protein containing a CARD; AAV, adeno-associated viruses; AMPK/Sirt-1, adenosine monophosphate-activated protein kinase/silent information regulator sirtuin 1; APN, adiponectin; BDNF, brain-derived neurotrophic factor; BV2, mouse microglial cell line; CAS-1, caspase 1; CDAD, cognitive dysfunction associated with depression; CUMS, chronic unpredictable mild stress; DCX, doublecortin; EPM, elevated plus maze; Hes1/Hes5/Hey1/Heyl, downstream genes of Notch signaling; HT22, mouse hippocampal HT22 cells; i.c.v., intracerebroventricular; IDO, indoleamine 2,3-dioxygenase; i.g., intragastric; IGF-1, insulin-like growth factor-1; IL-1β, interleukin one beta; IL-6, interleukin six; i.p., intraperitoneal; i.v., intraventricular; JNK, c-Jun N-terminal kinase; KAT, kynurenine aminotransferase; Ki67, cell proliferation marker Ki67; KO, knockout; LPS, lipopolysaccharide; mtROS, mitochondrial reactive oxygen species; mTOR, mammalian target of rapamycin; NGF, nerve growth factor; NICD, Notch intracellular domain; NMDA, N-methyl-D-aspartate; NLRP3, nucleotide-binding oligomerization domain-like receptor pyrin domain containing 3; Notch1, neurogenic locus notch homolog protein 1; OF, open field; p-AMPK, phosphorylated adenosine monophosphate-activated protein kinase; PD, Parkinson’s disease; p-mTOR, phosphorylated mammalian target of rapamycin; PPARα, peroxisome proliferator-activated receptor alpha; shRNA, short hairpin ribonucleic acid; STZ, streptozotocin; TNF-α, tumor necrosis factor alpha; VEGF-α, vascular endothelial growth factor alpha.

## 4. Discussion

In a study involving diabetic and depressive C57BL/6N mice, Zhao et al. [31] utilized AdipoRon at 20 mg/kg via intraperitoneal (i.p.) injection to induce antidepressant behaviors in the treated mice. The results demonstrated enhanced sucrose consumption, reduced immobility, increased swimming, and increased movement and distance in cross experiments. The pathways involved primarily anti-inflammatory avenues, including nucleotide-binding oligomerization domain-like receptor pyrin domain containing 3 (NLRP3) inhibition, apoptosis-associated speck-like protein containing a CARD (ASC) and interleukin one beta (IL-1β) reduction, an increase in the phosphorylated adenosine monophosphate-activated protein kinase (p-AMPK)/adenosine monophosphate-activated protein kinase (AMPK) ratio, and phosphorylated mammalian target of rapamycin (p-mTOR)/mammalian target of rapamycin (mTOR) inhibition in the mice brains, which led to hippocampal cell apoptosis inhibition, an increase in prefrontal cortex and hippocampus synapses, and dendritic spine neuronal density in the CA1 region enhancement.

Intranasal-administered AdipoRon was used to assess its antidepressant effects on a 6-OHDA-induced PD rat model. The results indicated that AdipoRon mitigated anxious and depressive-like behaviors by modulating the AMPK/silent information regulator sirtuin 1 (Sirt-1) signaling pathway and blocking NLRP3 inflammasome. Although these results are promising, the study includes a few limitations. Firstly, it is crucial to note that the researchers did not assess the precise pharmacokinetic parameters of AdipoRon administration while evaluating the results. These include half-life, brain penetration, and bioavailability. Moreover, the 6-OHDA rat PD model fails to accurately characterize the progressive nature of the human disease and the complex neuropathological parameters associated with the development of human PD [27]. Therefore, future research endeavors must prioritize using rat models of PD that more accurately reflect human development and its associated neuropsychiatric disorders. Intranasal administration may be a valuable route for enhancing the clinical use of AdipoRon, as it enables direct and rapid absorption due to the nasal mucosa’s rich vascularization. Intranasal administration can also maximize AdipoRon’s effectiveness, and given the proximity to the central nervous system, targeting the brain may be more feasible. It could also enable clinicians to not be preoccupied with the possible waste of resources due to the lack of exposure to gastric acid and the risk of first-pass hepatic passage. In this scenario, it is worth noting that intranasal drug administration is painless, readily available to all patients, and does not require intravenous access [34].

Liu et al. investigated the effects of AdipoRon administration in cultured hippocampal cells incubated with activated microglia against depression-like behaviors in mice. Their results indicated neuroprotection against depression-related alterations and depressive behaviors through the mitigation of NLRP3 inflammasome activation. Inhibiting the NLRP3 inflammasome also led to decreased mitochondrial reactive oxygen species (mtROS) accumulation and increased mitophagy in vitro, ultimately increasing the clearance of damaged mitochondria in the studied models of depression. The study has no evident methodological limitations, presents a significant time frame for AdipoRon administration in vivo, and cultured cells incubated with AdipoRon at different concentrations. The authors also used the following two controls: chronic unpredictable mild stress (CUMS) mice alone and CUMS mice treated with fluoxetine 20 mg/kg. Future research must investigate the effects of AdipoRon incubation or administration in other types of glial cells. Furthermore, future research should explore the molecular mechanisms underlying the antidepressant effects of AdipoRon at the cellular level using cell transfection and transgenic mice [28].

In a separate study, Formolo et al. sub-chronically administered AdipoRon to mice and evaluated its effects against depression-like behaviors. The increased adiponectin signaling achieved by sub-chronic AdipoRon treatment resulted in antidepressant- and anxiolytic-like effects. These effects were independent of changes in hippocampal structural and synaptic function [29]. However, the study has some limitations. The authors did not evaluate the molecular mechanisms involved. Although they used the knockdown of the N-methyl-D-aspartate (NMDA) receptor subunits GluN2A and GluN2B in CamKIIα-Cre mice, the antidepressant- and anxiolytic-like effects of AdipoRon were independently associated with this knockdown, which limits the generalizability of the observed results and precludes investigations into the related pathway. Additionally, seven days may not be sufficient to assess the penetration and bioavailability of AdipoRon in the central nervous system. Future research endeavors should prioritize the use of more comprehensive treatment protocols. Using recombinant adeno-associated virus (AAV) vectors to assess the pathophysiology of neurological diseases is convenient for several reasons: the models are flexible during design, and viruses allow for the manipulation of transgenes, promoters, regulatory elements, and reporter genes [35]. However, it is worth noting that these vectors may not be more effective than other vectors, such as lentiviral vectors, due to their need for a higher multiplicity of infection. Moreover, AAV often presents small genome sizes, which limits the size of the transgene that can be delivered to specific brain areas, ultimately leading to a reduced expression of genetic alterations in all brain areas affected during depressive disorders. Finally, particular routes of central administration can lead to brain stem injury, which could falsify results [36,37,38,39].

Despite the previous results, Li et al. reported both negative and positive effects of AdipoRon-based treatment against depression in a mouse model of LPS-induced depression-like behaviors. These authors demonstrated that while AdipoRon ameliorated the redox status of the treated mice, it also abolished the antidepressive-like behaviors presented by the mice, increasing immobility and decreasing sucrose preference. Although the results regarding depression were negative, the study demonstrated the anti-neuroinflammatory potential of AdipoRon. Treatment increased redox defensive mechanisms, upregulated nuclear factor erythroid 2-related factor 2 (Nrf2)/superoxide dismutase 2 (SOD2) expression, and modulated interleukin six (IL-6), transforming growth factor beta one (TGF−β1), interleukin ten (IL-10), and interleukin four (IL-4) expression [32]. Future research directions should aim to replicate AdipoRon’s results in chronic, prolonged treatments against neuroinflammation. In this study, the authors assessed AdipoRon’s effects after seven days of treatment, which may be considered undersized for inflammation amelioration and potentially lead to antidepressive behaviors. Chronically diminished neuroinflammation may lead to decreased depression-like behaviors since neuroinflammatory signaling is closely related to depression occurrence [40].

You et al. were the first to investigate the effects of the intraventricular cannulation and intracerebroventricular (i.c.v.) injection of AdipoRon against depression. These authors demonstrated that AdipoRon rescued hippocampal neurogenesis and ameliorated cognitive dysfunction associated with depression by principally upregulating Notch signaling and the genetic expression of key molecules [30]. The innovative study inaugurated a new AdipoRon administration directly into the central nervous system. However, it does not reflect translational research, since intraventricular and i.c.v. administrations are not clinically practical. Future research should investigate the administration of AdipoRon under prolonged and spaced treatment conditions, which would be more effective for clinical practice, such as in the spinal space. Neuraxial drug administration is a reality in anesthesia, as well as in the management of acute and chronic pain. However, caution is needed due to its possible off-target effects, which ultimately lead to neurotoxic effects. Delineation of the general AdipoRon pharmacology under neuraxial drug administration is warranted, especially regarding the specific aspects of epidural and intrathecal pharmacokinetics and pharmacodynamics, to improve efficacy and safety for possible new neuraxial AdipoRon treatments for depression [41].

Using long-term corticosterone treatment to induce depression in male mice, Nicolas et al. discovered new insights into AdipoRon’s effects against depression. These authors highlighted the antidepressant effects of AdipoRon by modulating neuroinflammation (IL-1β, IL-6, and TNF-α [tumor necrosis factor alpha] levels), restoring the physiological expressions of indoleamine 2,3-dioxygenase (IDO), kynurenine aminotransferase (KAT), and neurotrophic factors such as brain-derived neurotrophic factor (BDNF), vascular endothelial growth factor alpha (VEGF-α), insulin-like growth factor-1 (IGF-1), and nerve growth factor (NGF), and increasing the release and turnover of serotonin [33]. Although this study presented new molecular mechanisms associated with the antidepressant effects of AdipoRon, caution is warranted because of the sex-specific effects of the corticosterone model of depression. Additionally, it is unclear how to predict when the corticosterone model impairs positive valence behaviors, particularly if predicting the value of a treatment is necessary, as in the case of AdipoRon and its translational potential. Future research is needed to validate the model and replicate the results in different animal species; the use of transgenic mice may be necessary for this purpose. Table 2 summarizes the risk of bias assessment of the included in vivo studies based on SYRCLE’s risk of bias tool for animal studies [26].

Current antidepressant therapies primarily target the monoaminergic system. In this regard, selective serotonin reuptake inhibitors (SSRIs) and serotonin-norepinephrine reuptake inhibitors (SNRIs) are used to enhance the synaptic levels of serotonin and/or norepinephrine [42]. These are current first-line therapies for depressive disorders, often requiring several weeks to take effect, and are associated with significant adverse effects (e.g., sexual dysfunction, weight gain, and emotional blunting) [43,44]. Older antidepressant classes, including tricyclic antidepressants (TCAs) and monoamine oxidase inhibitors (MAOIs), possess broader mechanisms of action. Nevertheless, their use is limited by a higher risk of worsened adverse effects, including cardiotoxicity and dietary restrictions [45,46,47,48].

In contrast, newer approaches, including ketamine and its derivatives such as esketamine, act on the glutamatergic system, thereby antagonizing NMDA receptors, which results in rapid antidepressant effects through enhanced synaptogenesis and the modulation of BDNF. However, these are associated with dissociative symptoms and abuse liability, limiting their abundant clinical use [49,50]. Today, non-pharmacological treatment strategies, including cognitive behavioral therapy (CBT), are also a reliable option for treating depressive symptoms, often in association with antidepressant medication [51]. Novel non-pharmacological therapy strategies, including electroconvulsive therapy (ECT) [52] and repetitive transcranial magnetic stimulation (rTMS) [53], offer effective alternatives or adjuncts, particularly in treatment-resistant cases. Still, access and cost can be limiting factors.

In this scenario, AdipoRon emerges as a strategy to counteract depressive symptoms by targeting a completely different pathway, the AdipoR1 and AdipoR2 receptors, representing a mechanistically distinct antidepressant medication [14,33]. By stimulating AMPK signaling and reducing neuroinflammation, AdipoRon supports the upregulation of BDNF and mitochondrial function, which are potent antidepressive strategies [54,55,56,57]. Since AdipoRon mimics adiponectin signaling, it presents the utmost potential for benefiting patients with metabolic diseases with depression, including those co-living with depression and insulin resistance [31]. Unlike ketamine, based on the included studies that utilized AdipoRon treatment strategies for one week [29,32], two weeks [28,31], and three weeks [27,33], AdipoRon’s effects are more likely to be gradual. However, it potentially offers broader systemic benefits beyond those of antidepressants and may present fewer adverse events due to its selectivity [58,59]. These underscore AdipoRon’s potential to serve as a promising adjunct or alternative to existing treatments against depression, especially in metabolically vulnerable populations.

While the preclinical evidence on AdipoRon treating depression exists, a few gaps limit the translational potential of AdipoRon into well-structured clinical trials that culminate in clinical guidelines. Rodents possess distinct brain structures, neurological metabolism, and nervous receptor profiles that differ from those of humans [60,61,62]. Due to differences in metabolism, doses that are effective in rodents may be toxic or ineffective in humans. Also, drugs like AdipoRon may not cross the human blood–brain barrier as effectively as in rats [17]. In addition, depression in humans is more heterogeneous than that presented within preclinical models and is also often associated with chronic comorbidities [63,64,65]. In humans, unlike rats, the multifactorial nature of depression has been more well-funded, although recent studies have affirmed that physical or endocrine stress can trigger depression in animals [66,67]. However, preclinical findings, such as improvements in immobility time, are difficult to translate into clinical settings, as animal studies primarily focus on behavioral aspects [68]. In contrast, clinical trials focus more on patient-reported mood scales, which are more subjective but not less effective [69].

Therefore, despite the significant advances in neuroscience and psychopharmacology deriving from the use of AdipoRon therapies against depression in preclinical models, translating preclinical findings into practical clinical treatment strategies for depression remains a considerable challenge. Compounds that demonstrate robust antidepressant-like effects in animal models may not translate to efficacy or safety in human trials, creating translational gaps [70,71,72]. This disconnect arises from a range of factors, including species-specific differences in brain function, oversimplification of animal models, the lack of reliable biomarkers, and inconsistencies in pharmacokinetics between rodents and human subjects [67,73,74,75]. As a result, promising agents like AdipoRon with confirmed preclinical potential face significant hurdles in clinical applications. Bridging this gap requires rethinking the experimental designs often associated with this research, incorporating more clinically relevant models, translational endpoints, and rigorous pharmacological validation to improve the likelihood of successful translation from bench to bedside.

Firstly, advanced preclinical studies must be conducted. Rodents presenting chronic psychological stress with comorbid metabolic dysfunction must be utilized to assess AdipoRon’s antidepressant efficacy in a model that mimics human depression with metabolic comorbidity [76,77,78]. In this scenario, CUMS would be applied to high-fat-diet rats. Control, stress + vehicle, stress + AdipoRon at multiple doses, and stress + standard antidepressant (e.g., fluoxetine) groups would be utilized. The primary outcomes would be the results of the sucrose preference test, the forced swim test, and the novelty-suppressed feeding test. Glucose tolerance, body weight, and insulin sensitivity management would be secondary to these outcomes. Biochemical and molecular analyses, including BDNF, AMPK, and pro-inflammatory cytokine levels (such as IL-6 and TNF-α) in the hippocampus, would be considered tertiary outcomes. This study design would undoubtedly replicate real-world complexity and also compare AdipoRon therapy with established treatment strategies against depression.

Secondly, the penetration of AdipoRon through the blood–brain barrier would be more thoroughly discussed, as evidence underscores this potential, but with limited information on dynamic mechanisms [55]. Rodents must be given oral and i.p. deliveries of labeled AdipoRon. Plasma and brain tissue samples would be analyzed at multiple time points. Quantification of AdipoRon concentrations would be conducted numerous times during the experiment, including chronic, sub-chronic, and acute determinations following AdipoRon administration. Parameters like half-life, Tmax, brain/plasma AdipoRon ratio, and tissue distribution would successfully elucidate AdipoRon’s blood–brain barrier penetration, informing the most effective dosing for further studies. Thirdly, genetic or chemogenetic dissection studies could also be conducted. This would unveil the mechanistic validation of AdipoRon’s receptor targets, supporting pathways to clinical trials. In this context, AdipoR1/R2 knockout (KO) mice or mice with viral knockdown in the hippocampus would be utilized. Groups would involve AdipoRon, KO + AdipoRon, and KO vehicle. Behavioral tests would be conducted, including the forced swim and tail suspension tests, as well as molecular signaling studies involving AMPK and BDNF, and electrophysiology aspects in the hippocampus.

Following these preclinical endeavors, human trials would be more likely to occur, including microdosing studies at Phase Zero, safety and tolerability trials in Phase One, and Phase Two pilot studies assessing AdipoRon in patients with major depressive disorder and metabolic syndrome, conducted under the standards of clinical evaluations [79].

Converging evidence across our included studies underscores the potential of AdipoRon in inhibiting the NLRP3 inflammasome as a standard molecular mechanism underlying the antidepressant effects of this adiponectin receptor agonist. In the included preclinical models, AdipoRon treatment consistently suppressed the activation of the NLRP3 inflammasome and its downstream inflammatory mediators. AdipoRon inhibited hippocampal cell apoptosis, increased prefrontal cortex and hippocampal synapses, and enhanced dendritic spine neuronal density in the CA1 region via NLRP3 inhibition and a reduction in ASC and IL-1β levels [31]. Additionally, AdipoRon modulated anxious and depressive-like behaviors in a rat model of PD by principally blocking the NLRP3 activation and modulating its downstream [27]. Finally, AdipoRon also ameliorated neuroinflammation and depressive-like behaviors via improved microglia function through NLRP3 modulation [28]. Taken together, these findings highlight NLRP3 inflammasome inhibition as a convergent and critical pathway through which AdipoRon exerts its neuroprotective and antidepressant effects, reinforcing the inflammasome’s role as a promising therapeutic target with AdipoRon in mood disorders.

Comparative analysis reveals several critical concerns regarding AdipoRon administration routes, including i.p., intranasal, and i.c.v., with potential translational implications. The most commonly used route is i.p. injection. Via this route, multiple studies demonstrated antidepressant effects, which were consistent across the included studies in various models of depressive-like symptoms. On the other hand, intranasal administration was used in one study, showing antidepressant and anxiolytic effects at very low doses. This route is essential because it bypasses the blood–brain barrier and shows strong translational promise for neurodegenerative conditions in human subjects, resulting in less systemic exposure. Finally, i.c.v. injection showed more prominent outcomes related to neurogenesis and cognitive impairments. However, this method is invasive and does not meet the criteria for widespread use in clinical settings due to its characteristics. In summary, i.p. mimics systemic drug delivery and is more clinically feasible, but may present many peripheral effects. Intranasal offers non-invasive targeting of the central tissues of the encephalon, potentially improving drug delivery efficiency and reducing systemic side effects, which is valuable for neurodegenerative conditions related to depression. The i.c.v. injection, although it possesses less translational feasibility in humans due to invasiveness issues, can provide valuable mechanistic insights by directly targeting central structures. Therefore, our data suggest that the intranasal route could be prioritized in future translational studies, especially those related to central nervous system disorders. In contrast, the i.p. route remains useful for systemic metabolic stress models.

Regarding the included studies’ models, the included studies showed that AdipoRon exerted broad antidepressant efficacy across diverse depression models, particularly where metabolic dysfunctions and neuroinflammation are prominent. In this context, metabolic and stress-induced depression models show robust antidepressant effects. However, these can vary depending on the underlying pathology. Less efficacy in pure inflammatory models may be present in KO mice for adiponectin signaling, suggesting dependence on intact adiponectin signaling. Although the effects may vary depending on the model tested, common trends underlying neurogenesis, anti-inflammatory effects, and neurotransmitter regulation are observed.

Figure 3 depicts the main findings of the included studies.

Despite its significant findings, our review presents limitations. Firstly, it is solely based on preclinical studies. Although most of our results are derived from animal studies, the findings may not be generalizable to humans, as animal models do not fully replicate the pathophysiology of human depression or neuroinflammation. Secondly, our review includes preclinical research with multiple and differing study designs, models, AdipoRon doses, administration routes, and outcome measures. Third, there is a current lack of preclinical studies with long-term administration strategies, lacking long-term safety and efficacy assessments. Chronic depression requires long-term treatment, so short-duration findings may be of limited relevance. Fourthly, preclinical models may underreport or not evaluate potential side effects associated with AdipoRon administration. Therefore, side effect information may be overly optimistic given the therapeutic profile of AdipoRon. Fifthly, the current scope of evidence does not compare AdipoRon with established antidepressant medications, limiting the understanding of its relative efficacy and positioning within the broader treatment landscape. Finally, as in all review articles, publication bias for positive results in the included studies may be a concern. Therefore, the apparent efficacy of AdipoRon may be overestimated if negative or null results are underreported.

## 5. Conclusions

Analyzing the included studies in this comprehensive review shows that AdipoRon demonstrates contrasting effects against depression. However, most of the evidence underscores AdipoRon-based adiponectin replacement therapies as potential candidates for future treatments against this critical psychiatric condition. Future research endeavors must address several limitations. These have been tentatively highlighted throughout the manuscript to facilitate the future conduct of more detailed and informative preclinical studies, which will help to successfully translate the findings into clinical, randomized, and masked trials.

## Figures and Tables

**Figure 1 biomedicines-13-01867-f001:**
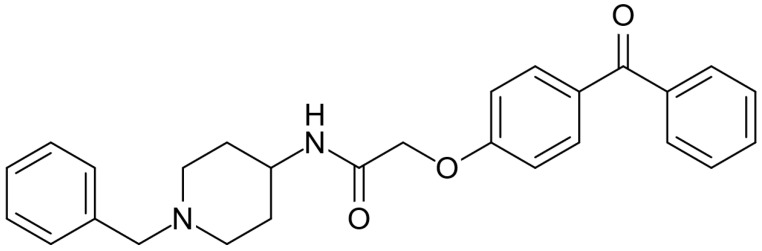
Molecular structure of AdipoRon.

**Figure 2 biomedicines-13-01867-f002:**
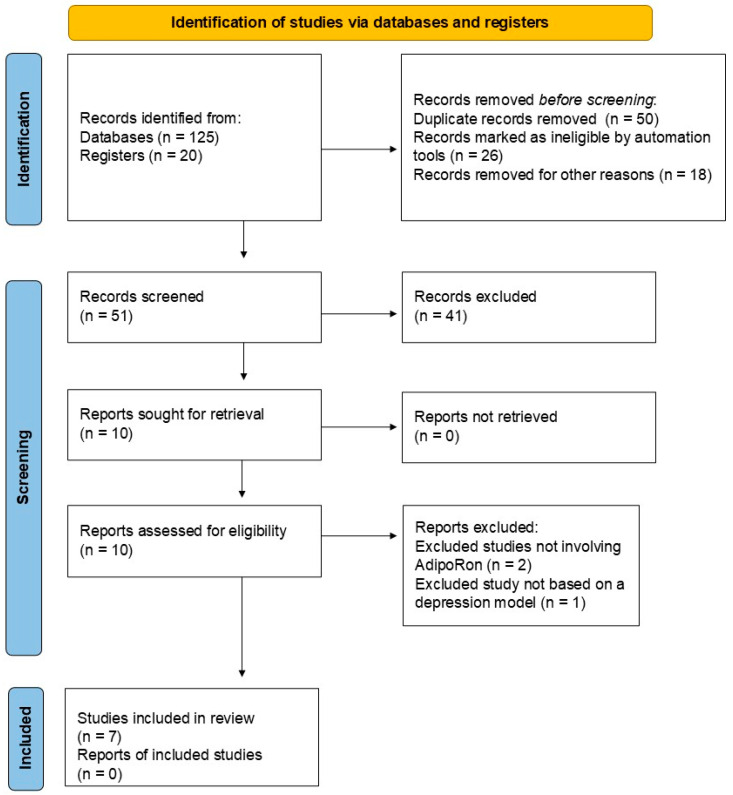
Literature search report.

**Figure 3 biomedicines-13-01867-f003:**
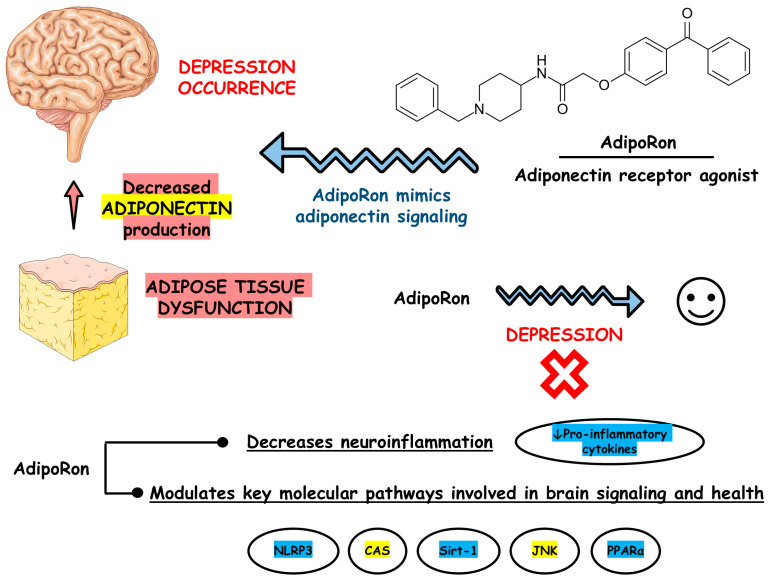
Main findings based on the included studies. Abbreviations: CAS, caspase; JNK, c-Jun N-terminal kinase; NLRP3, nucleotide-binding oligomerization domain-like receptor pyrin domain containing 3; PPARα, peroxisome proliferator-activated receptor alpha; Sirt-1, silent information regulator sirtuin 1.

**Table 2 biomedicines-13-01867-t002:** Animal preclinical studies risk of bias assessment based on SYRCLE’s risk of bias tool for animal studies [26].

References	D1	D2	D3	D4	D5	D6	D7	D8	D9	D10	Overall
Zhao et al. [31]							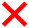				
Azizifar et al. [27]											
Liu et al. [28]											
Formolo et al. [29]											
Li et al. [32]	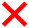	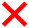		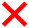	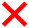	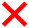	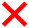				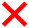
You et al. [30]											
Nicolas et al. [33]											

**Abbreviations:**

, Low risk of bias; 
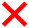
, High risk of bias; 

, Unclear risk of bias; D1, Sequence generation; D2, Baseline characteristics; D3, Allocation concealment; D4, Random housing; D5, Blinding of caregivers; D6, Random outcome assessment; D7, Blinding of outcome assessment; D8, Incomplete outcome data; D9, Selective outcome reporting; D10, Other bias.

## Data Availability

Not applicable.

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
