# Peer review of "AdipoRon as a Novel Therapeutic Agent for Depression: A Comprehensive Review of Preclinical Evidence"

_biomedicines, 2025, doi:10.3390/biomedicines13081867_

Round 1

Reviewer 1 Report

Comments and Suggestions for Authors

Thank you for the opportunity to review this relevant manuscript, which has potential clinical implications.
I have a few suggestions to improve the manuscript before it can be considered for publication:

Minor Comments:

  1. In theAbstract, please consider adding “to the best of our knowledge” to the sentence: “No comprehensive review has addressed these results.”
  2. In section1. Database Search, please describe the search strategy used for each database, including the total number of manuscripts retrieved and how many were ultimately included in the review.
  3. In section4. Data Extraction, please specify the percentage of agreement among authors during the data extraction process.

Major Comments:

The Discussion section largely reiterates the results without sufficient synthesis or exploration of mechanistic insights, clinical relevance, or future directions. I suggest the following improvements:

  1. Compare AdipoRon with other antidepressant strategies.
  2. Discuss the translational gap between preclinical and clinical research.
  3. Propose specific experimental designs for future studies.
  4. Add a paragraph discussing the limitations of the study at the end of the discussion section.

The manuscript shows a 31% similarity index. I recommend verifying the text to avoid potential plagiarism.

Author Response

RESPONSE TO REVIEWERS' COMMENTS

Manuscript number: biomedicines-3763659 ― Biomedicines (MDPI)

"Exploring the Effects of AdipoRon, an Adiponectin Receptor Agonist, Against Depression ― A Comprehensive Review"

The authors of this document wish to express their deepest gratitude to the Editor-in-Chief and the Reviewer for their thorough and insightful evaluation of our manuscript. Their expert feedback has been invaluable in enhancing the quality of our work. We have carefully considered and diligently implemented each suggestion, which has significantly improved the manuscript. We have made substantial revisions to address the points raised. These noteworthy changes are marked mainly with YELLOW-highlighted text throughout the document for ease of reference. A note will be provided for the referee's attention for corrections highlighted in a different color. Additionally, we have prepared a detailed and comprehensive response to each comment and suggestion. This response is organized in a "point-by-point" format below, ensuring that every concern has been thoroughly addressed and explained. We sincerely appreciate the time and effort invested by the Editor-in-Chief and the Reviewer, and we believe their contributions have significantly strengthened the final version of our manuscript.

REVIEWER #1

General comment

Thank you for the opportunity to review this relevant manuscript, which has potential clinical implications. I have a few suggestions to improve the manuscript before it can be considered for publication.

General response

Dear Erudite Reviewer, thank you for taking the time to revise our manuscript and allowing us to improve based on your valuable comments and suggestions. After addressing all your comments and suggestions regarding our manuscript text, we are confident that a significantly enhanced manuscript version has emerged. We are excited to resubmit the modified version for your perusal and reevaluation. Thank you for your brilliant insights, essential contributions, and feedback. You do have an eye for improvement. As a gesture of our utmost respect for you, we would like to provide you with a detailed and comprehensive point-by-point response to your comments below. Thank you once again for your time and patience in revising our article.

Comment #1

In theAbstract, please consider adding “to the best of our knowledge” to the sentence: “No comprehensive review has addressed these results.”

Response

Dear Erudite Reviewer, thank you for providing this critical feedback about our manuscript. I appreciate your commitment to ensuring that our submission is as accurate as possible, in line with the publication standards of this crucial journal, Biomedicines. We included the sentence you suggested in Lines 32-33 on Page 1 of the revised manuscript document. Thank you for your precious input. Our manuscript has been significantly improved based on your comments and suggestions.

Comment #2

In section1. Database Search, please describe the search strategy used for each database, including the total number of manuscripts retrieved and how many were ultimately included in the review.

Response

Dear Erudite Reviewer, we appreciate this comment, which significantly enhances our manuscript’s quality and readability. Therefore, we included Lines 200-212 on Page 5 of the revised manuscript document, delving into our study selection process across reputable databases. We identified 125 records from various sources: PubMed (75), Scopus (30), Web of Science (10), Google Scholar (10), and registers (20). After screening, we excluded 50 duplicates, 26 ineligible samples, and 18 for other reasons (17 were not in English, and 1 was not published within the last ten years). This left 51 records for inclusion screening, resulting in 41 exclusions, mainly reviews and non-experimental papers. We retrieved 10 studies, all of which were obtained and assessed for eligibility, resulting in 7 studies included in the final analysis. Three studies were excluded: two lacked AdipoRon-based therapies, and one had no depression models.

We believe that this additional content was invaluable in determining the quality of our reporting findings. The included studies are summarized in Table 1, which depicts the findings on AdipoRon’s impact on depression. Once again, thank you for your commitment to ensuring that our manuscript meets the highest standards. We appreciate your willingness to improve our manuscript accordingly.

Comment #3

In section 4. Data Extraction, please specify the percentage of agreement among authors during the data extraction process.

Response

Dear Erudite Reviewer, thank you for providing feedback on our data extraction process. We appreciate your willingness to make our article as transparent as possible. Therefore, we included Lines 175-177 on Page 4 of the revised manuscript document. Any disagreements that arose were resolved by a third author, V.D.R. However, there were no disputes among the authors during the process of identifying and selecting the studies that fulfilled our inclusion criteria. We believe that this fact represents our utmost criteria in ensuring that our included studies are of the utmost importance for our analysis, and that our inclusion criteria are accurate and reflect the current state-of-the-art regarding AdipoRon therapy against depression in preclinical models.

            Again, thank you for everything! Your critical comments have been invaluable in helping us refine our manuscript for the better. We are thankful for the opportunity to communicate with such an esteemed reviewer.

Comment #4

The Discussion section largely reiterates the results without sufficient synthesis or exploration of mechanistic insights, clinical relevance, or future directions. I suggest the following improvements. Compare AdipoRon with other antidepressant strategies.

Response

Dear Erudite Reviewer, thank you for this precious comment. We acknowledge your attention to detail and eye for improvement, and agree with you that adding some comparisons between AdipoRon and other antidepressant actions would provide a notable analytical aspect to our manuscript. Therefore, we included Lines 370-402 on Pages 11-12, delving into these intricacies. Current antidepressants mainly target the monoaminergic system (e.g., SSRIs, SNRIs), but have delayed effects and notable side effects. Older drugs like TCAs and MAOIs are less commonly used due to higher risks. Newer treatments like ketamine act rapidly via glutamatergic pathways but come with dissociative effects and abuse risk. Non-drug therapies (e.g., CBT, ECT, rTMS) offer alternatives, though access and cost can be barriers. AdipoRon offers a novel approach by targeting adiponectin receptors (AdipoR1/R2), thereby enhancing AMPK signaling, BDNF, and mitochondrial function, while reducing inflammation. It may work more gradually but has fewer side effects and potential benefits for patients with depression and metabolic disorders (e.g., obesity, insulin resistance), making it a promising alternative or adjunct to current therapies.

            Again, thank you for your eye for improvement. It is an honor for us to communicate with you and correct our manuscript based on your critical comments.

Comment #5

Discuss the translational gap between preclinical and clinical research.

Response

Dear Erudite Reviewer, thank you for this important comment and critical suggestions. You are entirely correct, and we agree with you that improving our manuscript in this regard would undoubtedly enhance our analysis. Therefore, we included Lines 403-416 on Page 12, delving into the translational gaps between preclinical and human research. While there is preclinical evidence supporting AdipoRon for treating depression, several gaps limit its potential for clinical trials and guidelines. Rodents have distinct brain structures and metabolic profiles compared to humans, meaning that effective doses in rodents may not be effective or could be toxic in humans. Additionally, AdipoRon may not cross the human blood-brain barrier as effectively as it does in rats. Human depression is more heterogeneous and often linked with chronic comorbidities, making it multifactorial compared to rodent models. Lab animals also lack a history of life stress and medication, rendering many preclinical findings non-replicable in humans. This disparity makes translating improvements from animal studies to clinical settings challenging, as animal research focuses on behavior while clinical trials rely on patient-reported mood scales.

            We believe that our text has been significantly improved following these additions. We appreciate your commitment to ensuring that our manuscript is of the highest quality. Thank you for your attention to detail and eye for improvement. It is a true honor to communicate with you.

Comment #6

Propose specific experimental designs for future studies.

Response

Dear Erudite Reviewer, thank you for addressing this paramount concern with us. We appreciate your attention to detail and look forward to improvement. We thank you for your commitment to ensuring that our manuscript is as good as it can be. We agree with you that discussing possible research designs to validate the translational potential of AdipoRon into clinical trials would enhance our analysis. Therefore, we included Lines 417-461 on Pages 12-13, exploring the rationale behind conducting additional preclinical and novel human trials. Despite notable progress in understanding AdipoRon's antidepressant potential in preclinical models, translating these findings into effective clinical treatments for depression remains challenging due to species differences, oversimplified models, and pharmacokinetic inconsistencies. To address these translational gaps, future research must adopt more clinically relevant approaches, such as using rodent models that mimic human depression with metabolic comorbidities, assessing behavioral and molecular outcomes, and comparing AdipoRon with standard antidepressants. Additionally, thorough evaluation of AdipoRon's blood-brain barrier penetration and dosing through pharmacokinetic studies, along with mechanistic validation using genetic or chemogenetic tools, is essential. These steps would lay the groundwork for clinical trials, beginning with microdosing and safety studies, and ultimately progressing to efficacy trials in patients with depression and metabolic syndrome.

            Again, thank you for everything. We appreciate your help and guidance, and eagerly await a positive response regarding the version of our manuscript. It is a true honor to revise our manuscript in response to your valuable comments and suggestions.

Comment #7

Add a paragraph discussing the limitations of the study at the end of the discussion section.

Response

Dear Erudite Reviewer, thank you for this comment. We understand the importance of acknowledging the limitations of our review. Therefore, we included Lines 512-528 on Page 15 of the revised manuscript discussing the limitations of our review. It relies solely on preclinical studies, mostly in animals, which may not fully reflect human depression or neuroinflammation. The studies vary widely in design, dosage, and methods, and the long-term effects of AdipoRon remain largely unexplored. Side effects may be underreported, and comparisons with existing antidepressants are lacking. Additionally, publication bias may have inflated the perceived effectiveness of AdipoRon.

            We appreciate the comment you provided, which has enhanced the quality and readability of our manuscript. We appreciate your willingness to improve our presentation, and look forward to your positive response. Thank you for everything!

Comment #8

The manuscript shows a 31% similarity index. I recommend verifying the text to avoid potential plagiarism.

Response

Thank you for bringing this matter to our attention. We have thoroughly reviewed the similarity report and conducted a detailed assessment of the overlapping content. In response, we have carefully revised the manuscript to significantly reduce the similarity index by rephrasing, restructuring, and properly citing relevant sections. These revisions were made with careful attention to maintaining the accuracy, clarity, and scientific integrity of the original work. We are confident that the updated version now fully adheres to academic and ethical standards regarding originality and proper attribution.

I, the corresponding author of the manuscript "Exploring the Effects of AdipoRon, an Adiponectin Receptor Agonist, Against Depression ― A Comprehensive Review" under the assigned ID biomedicines-3763659, on behalf of my coauthors, once again extend my heartfelt gratitude to the knowledgeable Editor-in-Chief and reviewers for their time and expertise in revising our manuscript. After we addressed their constructive and refined feedback and suggestions, a significantly improved manuscript version emerged. Undoubtedly, their insightful suggestions and feedback have significantly enhanced the quality of our manuscript. We respectfully are at the disposal of the Editor-in-Chief and the Reviewer to address any additional suggestions regarding our publication. If you are satisfied with our newly refined and significantly improved version, we look forward to the acceptance of our article for publication in this prestigious journal, Biomedicines. Thank you once again for your time and expertise.

Reviewer 2 Report

Comments and Suggestions for Authors

Reviewer report

The authors reviewed the effects of AdipoRon, an adiponectin receptor agonist, against depression. The review synthesizes findings from seven preclinical studies, summarizing their models, interventions, and mechanistic pathways. The central argument is that while evidence shows some contrasting effects, AdipoRon demonstrates significant potential, primarily through its anti-neuroinflammatory and neuroprotective actions (e.g., inhibiting the NLRP3 inflammasome, modulating AMPK/Sirt-1 signalling).

Major comments

  1. PRISMA-The authors state they conducted a comprehensive literature search across multiple databases. For a review claiming a systematic approach, it is standard and essential practice to include a PRISMA (Preferred Reporting Items for Systematic Reviews and Meta-Analyses) flow diagram. This diagram would visually depict the entire search and screening process: the number of records identified, duplicates removed, records screened, full-text articles assessed for eligibility, and the final number of studies included in the review, with reasons for exclusion at each stage. This is crucial for the transparency and reproducibility of the search process.
  2. Assessment/Risk of Bias Analysis: The "Quality Assessment" section (2.5) describes the process of assessment in general terms but does not specify which standardized tool was used. For preclinical animal studies, a tool such as the SYRCLE (Systematic Review Centre for Laboratory Animal Experimentation) risk of bias tool, or at minimum a detailed, pre-specified checklist, should be used. The results of this assessment (e.g., a summary graph or table showing the risk of bias for each included study across different domains like randomization, blinding, etc.) should be presented.
  3. Discussion: The discussion section is well-written but is structured as a study-by-study summary. While the critical evaluation of each study is a strength, the review would be significantly improved by including a more integrated synthesis of the findings. For example, the authors could add paragraphs that synthesize the evidence thematically:
    • A section discussing the convergent evidence on key molecular pathways (e.g., "The Role of NLRP3 Inflammasome Inhibition Across Studies").
    • A comparative analysis of the different administration routes used (i.p., intranasal, i.c.v.) and their implications for translational potential.
    • A synthesis of the different depression models used and whether AdipoRon's efficacy varies between them (e.g., metabolic vs. stress-induced vs. neurodegenerative models).
    • This would elevate the discussion from a series of summaries to a true synthesis of the current state of knowledge.

Minor comments

  • PICO Framework: In the "Database Search" section (2.1), the authors mention using the PICO framework. It would be beneficial to explicitly define the population, intervention, comparison, and outcomes that guided the search strategy to further improve clarity.
  • Table 1: This table is excellent but contains minor typographical errors.
    • In the row for [30], "immobility" should be "immobility".
    • In the same row, "mice' brains" should be "mice's brains" or "mouse brains".
    • In the row for [27], "Additionally" should be "Additionally".
    • In the row for [32], "impacted a depression-like state" contains an incorrect accent (dépression). This should be "depression".
  • Figure 2: This is a very effective summary figure. For clarity, the key ("↓, decrease") is slightly redundant, as the down-arrow symbol is universally understood. The authors could consider simplifying this.
  • Page 7, lines 239-242: The authors make an excellent point about intranasal administration. This section is very strong.
  • Page 9, lines 304-306: The point about the lack of translational potential for intraventricular administration is crucial and well-articulated.

Author Response

RESPONSE TO REVIEWERS' COMMENTS

Manuscript number: biomedicines-3763659 ― Biomedicines (MDPI)

"Exploring the Effects of AdipoRon, an Adiponectin Receptor Agonist, Against Depression ― A Comprehensive Review"

The authors of this document wish to express their deepest gratitude to the Editor-in-Chief and the Reviewer for their thorough and insightful evaluation of our manuscript. Their expert feedback has been invaluable in enhancing the quality of our work. We have carefully considered and diligently implemented each suggestion, which has significantly improved the manuscript. We have made substantial revisions to address the points raised. These noteworthy changes are marked mainly with YELLOW-highlighted text throughout the document for ease of reference. A note will be provided for the referee's attention for corrections highlighted in a different color. Additionally, we have prepared a detailed and comprehensive response to each comment and suggestion. This response is organized in a "point-by-point" format below, ensuring that every concern has been thoroughly addressed and explained. We sincerely appreciate the time and effort invested by the Editor-in-Chief and the Reviewer, and we believe their contributions have significantly strengthened the final version of our manuscript.

REVIEWER #2

General comment

The authors reviewed the effects of AdipoRon, an adiponectin receptor agonist, against depression. The review synthesizes findings from seven preclinical studies, summarizing their models, interventions, and mechanistic pathways. The central argument is that while evidence shows some contrasting effects, AdipoRon demonstrates significant potential, primarily through its anti-neuroinflammatory and neuroprotective actions (e.g., inhibiting the NLRP3 inflammasome, modulating AMPK/Sirt-1 signalling).

General response

Dear Erudite Reviewer, thank you for taking the time to revise our manuscript and allowing us to improve based on your valuable comments and suggestions. After addressing all your comments and suggestions regarding our manuscript text, we are confident that a significantly enhanced manuscript version has emerged. We are excited to resubmit the modified version for your perusal and reevaluation. Thank you for your brilliant insights, essential contributions, and feedback. You do have an eye for improvement. As a gesture of our utmost respect for you, we would like to provide you with a detailed and comprehensive point-by-point response to your comments below. Thank you once again for your time and patience in revising our article.

Comment #1

PRISMA-The authors state they conducted a comprehensive literature search across multiple databases. For a review claiming a systematic approach, it is standard and essential practice to include a PRISMA (Preferred Reporting Items for Systematic Reviews and Meta-Analyses) flow diagram. This diagram would visually depict the entire search and screening process: the number of records identified, duplicates removed, records screened, full-text articles assessed for eligibility, and the final number of studies included in the review, with reasons for exclusion at each stage. This is crucial for the transparency and reproducibility of the search process.

Response

Dear Erudite Reviewer, we appreciate this comment, which significantly enhances the quality and readability of our manuscript. Therefore, we included Lines 200-212 on Page 5 of the revised manuscript document, delving into our study selection process across reputable databases. We identified 125 records from various sources: PubMed (75), Scopus (30), Web of Science (10), Google Scholar (10), and registers (20). After screening, we excluded 50 duplicates, 26 ineligible samples, and 18 for other reasons (17 were not in English, and one was not published within the last ten years). This left 51 records for inclusion screening, resulting in 41 exclusions, mainly reviews and non-experimental papers. We retrieved 10 studies, all of which were obtained and assessed for eligibility, resulting in 7 studies included in the final analysis. Three studies were excluded: two lacked AdipoRon-based therapies, and one had no depression models. Figure 2 illustrates the literature search process and report. This figure can be found on Page 6 of the revised manuscript document. Additionally, its first mention can be found in Line 212 on Page 5, and its legend is in Line 214 on Page 6 of the revised manuscript document.

We believe that this additional content was invaluable in determining the quality of our reporting findings. The included studies are summarized in Table 1, which depicts the findings on AdipoRon’s impact on depression. Once again, thank you for your commitment to ensuring that our manuscript meets the highest standards. We appreciate your willingness to improve our manuscript accordingly.

Comment #2

Assessment/Risk of Bias Analysis: The "Quality Assessment" section (2.5) describes the process of assessment in general terms but does not specify which standardized tool was used. For preclinical animal studies, a tool such as the SYRCLE (Systematic Review Centre for Laboratory Animal Experimentation) risk of bias tool, or at minimum a detailed, pre-specified checklist, should be used. The results of this assessment (e.g., a summary graph or table showing the risk of bias for each included study across different domains like randomization, blinding, etc.) should be presented.

Response

Dear Erudite Reviewer, thank you for bringing this vital comment to my attention. We agree with you that standardizing our quality assessment tool would be of utmost importance to elevate the scientific rigor of our analysis. Therefore, we used SYRCLE's risk of bias tool for animal studies to assess the risk of bias of each animal study we included in our final analysis. Table 2, which presents the risk of bias assessment based on SYRCLE's risk of bias tool for animal studies, is located on Page XX of the revised manuscript document. Our methodology section has been updated in Lines 179-181 on Page 5 to depict the new quality tool we used. Table 2’s first mention can be found in Lines 361-363 on Page 11, and its caption can be found in Lines 365-368 on Page 11.

            Again, thank you for the opportunity to revise our manuscript in accordance with your valuable comments. We appreciate your willingness to provide essential recommendations to elevate our manuscript’s analysis. We appreciate your commitment to ensuring that our manuscript reaches the highest standards.

Comment #3

Discussion: The discussion section is well-written but is structured as a study-by-study summary. While the critical evaluation of each study is a strength, the review would be significantly improved by including a more integrated synthesis of the findings. For example, the authors could add paragraphs that synthesize the evidence thematically.

A section discussing the convergent evidence on key molecular pathways (e.g., "The Role of NLRP3 Inflammasome Inhibition Across Studies").

A comparative analysis of the different administration routes used (i.p., intranasal, i.c.v.) and their implications for translational potential.

A synthesis of the different depression models used and whether AdipoRon's efficacy varies between them (e.g., metabolic vs. stress-induced vs. neurodegenerative models).

This would elevate the discussion from a series of summaries to a true synthesis of the current state of knowledge.

Response

Dear Erudite Reviewer, thank you for this important suggestion. We agree with you that synthesizing this information would undoubtedly enhance the overall quality and presentation of our manuscript's findings. Therefore, we included Lines 462-506 on Pages 13-14 in the revised manuscript document to implement these needed modifications. AdipoRon consistently demonstrates antidepressant effects across preclinical models by inhibiting the NLRP3 inflammasome, thereby reducing inflammation, promoting neurogenesis, and enhancing synaptic function. It also modulates anxious and depressive behaviors through anti-inflammatory pathways, particularly involving microglia and the suppression of cytokines. Among delivery routes, intranasal administration shows strong translational potential due to its non-invasive nature and brain-targeting efficiency, while intraperitoneal (i.p.) remains useful for systemic models. Intracerebroventricular (i.c.v.) injections, although mechanistically informative, are clinically impractical. AdipoRon is especially effective in depression models tied to metabolic dysfunction and neuroinflammation, though its efficacy may vary depending on the model and requires intact adiponectin signaling. Overall, it holds promise as a neuroprotective antidepressant targeting inflammasome pathways.     

Thank you for the opportunity to revise our manuscript in accordance with your invaluable feedback. We are grateful for your willingness to offer essential recommendations that will enhance the analysis presented in our manuscript. We sincerely appreciate your commitment to ensuring that our work meets the highest academic standards.

Comment #4

PICO Framework: In the "Database Search" section (2.1), the authors mention using the PICO framework. It would be beneficial to explicitly define the population, intervention, comparison, and outcomes that guided the search strategy to further improve clarity.

Response

Dear Erudite Reviewer, thank you for providing this valuable comment, which has undoubtedly enhanced the overall quality and readability of our manuscript. We included the information regarding our PICO framework in Lines 136-149 on Page 4 of the revised manuscript document. We examined preclinical models of depression, including both cellular and animal studies (P). The intervention involved AdipoRon-based adiponectin replacement therapies (I), compared primarily to untreated or placebo groups, with some studies also comparing other antidepressants (C). Outcomes assessed included depression-like behaviors, neuroinflammation, cognitive function, and related molecular and metabolic effects (O). Research Question: In preclinical depression models, does AdipoRon treatment improve depression-related outcomes compared to no treatment or placebo?

            We believe that our manuscript has been significantly enhanced after we included your suggestions into our text. We look forward to your positive response regarding our modified version, based on your valuable input. Thank you for everything!

Comment #5

Table 1: This table is excellent but contains minor typographical errors.

In the row for [30], "immobility" should be "immobility".

In the same row, "mice' brains" should be "mice's brains" or "mouse brains".

In the row for [27], "Additionally" should be "Additionally".

In the row for [32], "impacted a depression-like state" contains an incorrect accent (dépression). This should be "depression".

Response

Thank you very much for your careful review and for bringing these typographical errors in Table 1 to our attention. We greatly appreciate your close reading and helpful observations. Based on your comments, we have corrected the following items. Please note that the reference numbers of the rows have been modified since additional references have been added to the manuscript.

In the first row of the table, the word "immobility" has been revised for consistency.

The phrase "mice' brains" has been corrected to "mouse brains" to reflect proper grammatical structure. Alternatively, “mouse brains” may be used interchangeably depending on context.

In the third row of the table, the word “Additionally” has been corrected.

In the seventh row of the table, we have removed the incorrect accent in the word "dépression" and replaced it with the correct form, "depression."

These revisions have been implemented in the updated version of Table 1, which now appears on Pages 7–8 of the manuscript. To assist in locating the changes, we have highlighted the relevant text in yellow.

We hope these corrections improve the clarity and accuracy of the table, and we thank you once again for your valuable input.

Comment #6

Figure 2: This is a very effective summary figure. For clarity, the key ("↓, decrease") is slightly redundant, as the down-arrow symbol is universally understood. The authors could consider simplifying this.

Response

We thank the reviewer for the positive feedback on Figure 3 and are pleased to hear that it is viewed as a practical summary. We also appreciate your thoughtful suggestion regarding the key, specifically the notation “↓, decrease.” We agree that the down-arrow symbol is widely recognized and that this clarification may be redundant. In response to your comment, we have revised the figure legend to simplify this element accordingly, thereby improving clarity and visual efficiency. Please find the revised legend for Figure 3 in Line 510 on Page 15 of the revised manuscript document. The legend is now “Figure 3. Main findings based on the included studies.”

Again, thank you for your attention to detail and eye for improvement! Your comments have significantly enhanced the quality and readability of our manuscript. Thank you for everything!

Comment #7

Page 7, lines 239-242: The authors make an excellent point about intranasal administration. This section is very strong.

Response

We sincerely appreciate the reviewer’s thoughtful and encouraging comment regarding our discussion of intranasal administration. We are very pleased to hear that this section was considered strong and that the point we raised resonated well. Positive feedback such as yours reinforces the value of including this discussion in our manuscript, and we thank you for acknowledging its relevance and clarity.

Comment #8

Page 9, lines 304-306: The point about the lack of translational potential for intraventricular administration is crucial and well-articulated.

Response

We sincerely thank the reviewer for highlighting our discussion on the lack of translational potential for intraventricular administration. We are grateful that you found this point to be both crucial and well-articulated. We believe it is essential to draw attention to these translational gaps to help guide future research toward more clinically feasible approaches. We greatly appreciate your recognition of this aspect of the manuscript.

I, the corresponding author of the manuscript "Exploring the Effects of AdipoRon, an Adiponectin Receptor Agonist, Against Depression ― A Comprehensive Review" under the assigned ID biomedicines-3763659, on behalf of my coauthors, once again extend my heartfelt gratitude to the knowledgeable Editor-in-Chief and reviewers for their time and expertise in revising our manuscript. After we addressed their constructive and refined feedback and suggestions, a significantly improved manuscript version emerged. Undoubtedly, their insightful suggestions and feedback have significantly enhanced the quality of our manuscript. We respectfully are at the disposal of the Editor-in-Chief and the Reviewer to address any additional suggestions regarding our publication. If you are satisfied with our newly refined and significantly improved version, we look forward to the acceptance of our article for publication in this prestigious journal, Biomedicines. Thank you once again for your time and expertise.

Round 2

Reviewer 1 Report

Comments and Suggestions for Authors

The authors have incorporated all the required modifications in accordance with my requests. However, the paragraphs between lines 391 and 461 lack references, which increases the risk of plagiarism or raises suspicions of AI-generated writing.

Author Response

RESPONSE TO REVIEWERS' COMMENTS

Manuscript number: biomedicines-3763659 ― Biomedicines (MDPI)

"AdipoRon as a Novel Therapeutic Agent for Depression: A Comprehensive Review of Preclinical Evidence"

The authors express their gratitude to the Editor-in-Chief and Reviewer for their insightful evaluation of our manuscript. Their feedback has been invaluable in enhancing our work, leading us to implement all suggestions and significantly improve the manuscript. Key revisions are marked in YELLOW for easy reference. We have also included a detailed, point-by-point response to each comment and suggestion. We greatly appreciate the time and effort of both the Editor-in-Chief and Reviewer, who have strengthened our final manuscript.

REVIEWER #1

General comment

The authors have incorporated all the required modifications in accordance with my requests.

General response

Dear Reviewer, thank you for revising our manuscript and providing valuable feedback. We have addressed all your comments and are excited to resubmit this improved version for your review. Your insights have greatly contributed to enhancing our work. Below, we provide a detailed point-by-point response to your suggestions. Thank you again for your time and patience.

Comment #1

However, the paragraphs between lines 391 and 461 lack references, which increases the risk of plagiarism or raises suspicions of AI-generated writing.

Response

Dear Erudite Reviewer, thank you for this comment. We understand the necessity of including references in these paragraphs. Therefore, we included 34 references between Lines 391-464 on Pages 11-13 in the revised manuscript document. We ensured the rigor and consistency in our writing by doing this, and we eagerly anticipate a positive response from you. Please note that, as per the nature of your previous comment in our manuscript, “Propose specific experimental designs for future studies” and the current lack of clinical validation based on our explanations and limitations raised by our in-depth analysis of the current issues and research gaps regarding AdipoRon strategies against depression, Lines 435-444 on Page 12 and Lines 447-460 on Page 13 cannot be referenced, since these are our propositions to advance knowledge in this critical area, which have been wrote in response to your previous comment. The added references are highlighted in yellow. Thank you for addressing these significant concerns with us. We believe that your critical comments, our distinct activity in resolving the issues you raised, and the addition of these references have richly enhanced our manuscript. We hope you can accept our manuscript for publication in this crucial journal.

I, the corresponding author of "AdipoRon as a Novel Therapeutic Agent for Depression: A Comprehensive Review of Preclinical Evidence" (ID: biomedicines-3763659), on behalf of my coauthors, extend my sincere gratitude to the Editor-in-Chief and reviewers for their valuable feedback. We have addressed their suggestions, resulting in a significantly improved manuscript. We are available for any further recommendations and look forward to the acceptance of our revised article in Biomedicines. Thank you for your time and expertise.

Reviewer 2 Report

Comments and Suggestions for Authors

The authors have made recommended corrections. However, minor corrections are still required.

Minor Suggestions 

  • Title: The title is good and descriptive. A slightly more assertive alternative could be: "AdipoRon as a Novel Therapeutic Agent for Depression: A Comprehensive Review of Preclinical Evidence." 

  • Discussion Flow: In the discussion, you address each study or group of studies sequentially, which is logical. You then synthesize these findings into thematic paragraphs (e.g., "Converging evidence," administration routes). This structure works very well.

  • Final Check: r Minor typos or grammatical errors should be checked.

Author Response

RESPONSE TO REVIEWERS' COMMENTS

Manuscript number: biomedicines-3763659 ― Biomedicines (MDPI)

"AdipoRon as a Novel Therapeutic Agent for Depression: A Comprehensive Review of Preclinical Evidence"

The authors express their gratitude to the Editor-in-Chief and Reviewer for their insightful evaluation of our manuscript. Their feedback has been invaluable in enhancing our work, leading us to implement all suggestions and significantly improve the manuscript. Key revisions are marked in YELLOW for easy reference. We have also included a detailed, point-by-point response to each comment and suggestion. We greatly appreciate the time and effort of both the Editor-in-Chief and Reviewer, who have strengthened our final manuscript.

REVIEWER #2

General comment

The authors have made recommended corrections. However, minor corrections are still required.

General response

Dear Reviewer, thank you for revising our manuscript and providing valuable feedback. We have addressed all your comments and are excited to resubmit this improved version for your review. Your insights have greatly contributed to enhancing our work. Below, we provide a detailed point-by-point response to your suggestions. Thank you again for your time and patience.

Comment #1

Title: The title is good and descriptive. A slightly more assertive alternative could be: "AdipoRon as a Novel Therapeutic Agent for Depression: A Comprehensive Review of Preclinical Evidence."

Response

Dear Erudite Reviewer, thank you for this important suggestion and critical improvement. We acknowledge the importance of a descriptive title for our manuscript, principally because it will be indexed in major databases, including PubMed. We accept your suggestion and have implemented the modification you provided in the manuscript. Therefore, our new title, as presented in Lines 2-3 on Page 1 of the revised manuscript document, is “ AdipoRon as a Novel Therapeutic Agent for Depression: A Comprehensive Review of Preclinical Evidence,” following your precious input. We believe our manuscript has been significantly improved based on your comments, and we appreciate the opportunity to communicate with you. Thank you again for your patience and guidance.

Comment #2

Discussion Flow: In the discussion, you address each study or group of studies sequentially, which is logical. You then synthesize these findings into thematic paragraphs (e.g., "Converging evidence," administration routes). This structure works very well.

Response

Dear Erudite Reviewer, thank you for this comment and for acknowledging our logical structure of text flow. We appreciate this comment and look forward to your positive response in the revised version, as well as to the approval of our manuscript for publication in the esteemed journal Biomedicines. Most of our manuscript’s strengths come from you, our Esteemed Reviewers. Therefore, we would like to thank you for everything.

Comment #3

Final Check: r Minor typos or grammatical errors should be checked.

Response

Dear Erudite Reviewer, thank you for this comment. We appreciate the scope of your review, and we thank you for bringing to our attention the necessity to read the manuscript and correct any remaining typographical issues. Therefore, we critically read the manuscript and corrected the minor typographical issues we found in Line 30 on Page 1, Line 358 on Page 10, Line 359 on Page 10, and Line 361 on Page 10. We believe that our manuscript has been strengthened from these corrections, and we truly appreciate your commitment to ensuring that every corner of our manuscript is improved before final publication. Thank you for your guidance!

I, the corresponding author of "AdipoRon as a Novel Therapeutic Agent for Depression: A Comprehensive Review of Preclinical Evidence" (ID: biomedicines-3763659), on behalf of my coauthors, extend my sincere gratitude to the Editor-in-Chief and reviewers for their valuable feedback. We have addressed their suggestions, resulting in a significantly improved manuscript. We are available for any further recommendations and look forward to the acceptance of our revised article in Biomedicines. Thank you for your time and expertise.